# Mechanochemical-Assisted Extraction and Biological Activity Research of Phenolic Compounds from Lotus Seedpod (*Receptaculum Nelumbinis*)

**DOI:** 10.3390/molecules28247947

**Published:** 2023-12-05

**Authors:** Nina Bao, Jiajia Song, Xinyuan Zhao, Marwan M. A. Rashed, Kefeng Zhai, Zeng Dong

**Affiliations:** 1School of Biological and Food Engineering, Suzhou University, Suzhou 234000, China; ninabao@ahszu.edu.cn (N.B.); 18256237724@163.com (J.S.); 18356603624@139.com (X.Z.); marwanrashed6@ahszu.edu.cn (M.M.A.R.); dongzeng@ahszu.edu.cn (Z.D.); 2Engineering Research Center for Development and High Value Utilization of Genuine Medicinal Materials in North Anhui Province, Suzhou University, Suzhou 234000, China; 3College of Materials and Chemical Engineering, Southwest Forestry University, Kunming 650224, China

**Keywords:** mechanochemical-assisted extraction, phenolic compounds, response-surface methodology, UPLC-Triple-TOF/MS, biological activities

## Abstract

To explore the feasibility of the mechanochemical-assisted extraction (MCAE) of phenolic compounds from lotus seedpod (*Receptaculum Nelumbinis*), a single-factor experiment combined with response-surface methodology (RSM) was used to optimize the extraction process. The results showed the optimal extraction conditions as follows: Li_2_CO_3_ as a solid reagent (25%), an extraction time of 80 min, liquid/solid ratio of 42.8 mL/g, and extraction temperature of 80.7 °C; and the maximum value of total phenolic content (TPC) was 106.15 ± 1.44 gallic acid equivalents (GAE)/g dry weight (DW). Additionally, the 2,2-Diphenyl-1-picrylhydrazyl (DPPH), 2,2′-azinobis (3-ethylbenzothiazoline-6-sulfonic acid) (ABTS), and ferric reducing antioxidant power (FRAP) were 279.75 ± 18.71, 618.60 ± 2.70, and 634.14 ± 7.17 µmol TE/g, respectively. Ultra-high pressure liquid chromatography combined with triple-time-of-flight mass spectrophotometry (UPLC-Triple-TOF/MS) analysis identified eight phenolic compounds mainly consisting of polyphenols and flavonoids. Moreover, the phenolic compounds showed potent inhibitory effects on both α-amylase and α-glucosidase, with inhibition rates of over 80%. Furthermore, the results showed different degrees of inhibition activity against *Bacillus subtilis*, *Staphylococcus aureus,* and *Escherichia coli*, among which the inhibitory effect on the growth of *B. subtilis* was the best. This paper shows that the phenolic compounds have good biological activities, which provides a reference for the further exploitation of LSP.

## 1. Introduction

Lotus (*Nelumbo nucifera* Gaertn.), a perennial aquatic plant belonging to the Nelumbonaceae family, is widely distributed and commonly cultivated in Asia and the US, and is traditionally utilized as a staple food and a medicine herb in China [1]. Lotus seedpod (LSP), an inedible by-product of the processing of lotus (*Nelumbo nucifera* Gaertn.), is considered agro-industrial waste in various industries and daily domestic life [1]. Not only does the production of this by-product result in an environmental burden, but it also leads to a massive waste of resources [2]. Lotus seedpod has been used as a conventional Chinese herbal medicine to treat blood stasis, as a disinfectant, to dispel dampness, to cease bleeding, and so on [2,3]. Moreover, researchers have demonstrated that lotus seedpod has the following activities: antioxidant [4], anti-inflammatory [5], anti-microorganisms [6], antiproliferative [7], and anti-gout [2]. These properties are mainly because of the rich phenolic compounds [4] of LSP, which could lead to it becoming a new source of phenolic compounds [8]. Therefore, it is advantageous to recover phenolic compounds from LSP.

In recent years, various extraction techniques used to recover phenolic compounds from plant materials have attracted more attention. These techniques were selected according to the extraction yield and production cost. However, techniques used to extract lotus seedpods are conventional, usually take a long time, and cannot exploit the enormous potential of the substrate. Additionally, organic solvents are used in the conventional extraction process, which are flammable, explosive, and toxic, leading to environmental pollution and the greenhouse effect [9]. Therefore, innovating a “green-based extraction process” by applying eco-friendly solvents is urgent in light of current technological developments.

Mechanochemical-assisted extraction is an innovative pre-extraction technology that has been successfully developed and widely applied to obtain natural products, such as antioxidant phenolic compounds [10,11], polysaccharides [12], and triterpene acids [13]. The mechanism of MCAE can be described as mechanical force (usually grinding in a ball mill), leading to the chemical and physicochemical transformations between the plant matrix and the solid regent (usually carbonated salts) by rupturing the cell wall that increases the total contact surface area, which improves the water solubility of the target substances [10,14].

The present study aimed to develop an efficient MCAE process and assess the bioactivities of the LSP phenolic compounds. In addition, the MCAE conditions, such as the solid reagent, liquid/solid ratio, extraction time, and extraction temperature, were conducted to assess the compounds individually, which were subsequently optimized using response-surface methodology. UPLC-Triple-TOF/MS was performed to identify and characterize the LSP phenolic compounds under the optimized conditions. Furthermore, the in vitro antioxidant, enzyme inhibitory, and antibacterial activities of phenolic compounds under optimized conditions were determined.

## 2. Results and Discussion

### 2.1. Selection of Solid Reagents

Solid reagents played an essential role in mechanochemical-assisted extraction processes. The forms of phenolic compounds in the plant materials restricted the type and concentration of solid reagents [11]. Eight different solid reagents, i.e., blank (no solid reagent), BaCO_3_, Na_2_B_4_O_7_·10H_2_O, K_2_CO_3_, Li_2_CO_3_, NaHCO_3_, Na_2_CO_3_, CaCO_3_, and CoCO_3_ were selected. The ability of each reagent to recover phenolic compounds was assessed under the conditions of a solid reagent amount of 25%, extraction time of 30 min, liquid/solid ratio of 20 mL/g, extraction temperature of 30 °C, and a magnetic stirring speed of 300 r/min. To assess MCAE’s effectiveness, a comparison was made with conventional extraction using 60% ethanol under the same extraction conditions. In comparison to water or 60% ethanol, the TPC values obtained using MCAE with Na_2_B_4_O_7_·10H_2_O, Li_2_CO_3_, CaCO_3_, and CoCO_3_ were higher, notably with Na_2_B_4_O_7_·10H_2_O and Li_2_CO_3._ Conversely, NaHCO_3_, K_2_CO_3_, and BaCO_3_ yielded lower values, making them less ideal. The value of TPC obtained using MCAE-Li_2_CO_3_ was 87.24 ± 1.98 mg GAE/g DW, which was significantly higher than Na_2_B_4_O_7_·10H_2_O (66.65 ± 0.73 mg GAE/g DW) or Na_2_CO_3_ (49.30 ± 0.63 mg GAE/g DW (Figure 1A), not to mention water (36.63 ± 0.82 mg GAE/g DW) or 60 % ethanol (41.26 ± 3.29 mg GAE/g DW). This phenomenon can be attributed to two factors: firstly, Li_2_CO_3_ is more effective in disintegrating and destroying the cell wall of LSP tissues [15,16]; secondly, phenolic compound salts may be more readily formed under Li_2_CO_3_ than Na_2_B_4_O_7_·10H_2_O or Na_2_CO_3_ mechanochemical pretreatment, which can improve the water solubility of phenolic compounds [15,17]. Thus far, MCAE has been widely used to extract bioactive components, including phenolics, from plant materials. A study reported that phenolic compounds were extracted from *Laurus nobilis* by using MCAE-Li_2_CO_3_ compared with conventional methods [10]. The value of phenolic compounds by these two methods is similar. However, the extraction time of MCAE-Li_2_CO_3_ is reduced more than ten times compared with the conventional method. Xu et al [18] reported their work in reducing energy and solvent consumption, and potentially achieving complete utilization of plant materials, which was used to demonstrate that mechanochemical-assisted extraction is an environmentally friendly and effective extraction method.

### 2.2. Optimization of Phenolic Compounds Extraction Process

#### 2.2.1. Single-Factor Experiments

The influence of different factors on TPC, including the extraction time, liquid/solid ratio, and extraction temperature, were estimated. As shown in Figure 1B, TPC increased when the extraction time increased from 10 to 70 min. However, a further increase in the extraction time did not benefit TPC. This is because a prolonged extraction time within a certain range might lead to cell wall rupture, thus promoting the total solubility of TPC. Obviously, a longer extraction time cannot obtain more TPC. Therefore, the best extraction time for the highest TPC was suggested at 70 min. For the liquid/solid ratio, the yield of TPC increased when the liquid/solid ratio increased from 20 mL/g to 40 mL/g. In contrast, a significant decrease in TPC was seen when the liquid/solid ratio increased to 50 mL/g. This could be due to the increased volume of the solution and enhanced mass transfer. However, the continuing increase in the solution volume leads to the reduction in the concentration gradient [19]. Thus, 40 mL/g was considered as the optimum liquid/solid ratio. The extraction temperature also affected TPC significantly, where an increase in the extraction temperature from 30 to 80 °C increased TPC (Figure 1D). Still, the yield then decreased when the extraction temperature was further increased. This phenomenon is attributed to the mass transfer increase under a certain threshold; once exceeded, the thermal instability of TPC occurs, leading to the degradation of TPC [20]. Hence, the most suitable extraction temperature was 80 °C for the highest yield of TPC. However, based on the central composite design (CCD), 75 °C was considered the optimum extraction temperature. According to the results obtained, the center level of CCD was set as 70 min of extraction time, 40 mL/g of liquid/solid ratio, and a 75 °C extraction temperature. The results were further verified in interactive experimental design.

#### 2.2.2. Interactive Effect on TPC Using MCAE

Table 1 summarizes the central composite design alongside experimental values and predicted TPC extracted from LSP using MCAE. The predicted values of TPC were calculated using the polynomial equation as follows:TPC = 107.58 + 2.61X_1_ + 10.98X_2_ + 7.93X_3_ − 4.74X_1_X_2_ + 3.71X_1_X_3_ + 0.91X_2_X_3_ − 7.98X_1_^2^ − 12.04X_2_^2^ − 10.53X_3_^2^(1)

The quadratic polynomial regression model of response surface with regression coefficient values is listed in Table 1, which evaluates the significant terms, such as the linear, quadratic, and cross product. The *p*-value of the “Model” is <0.0001, which demonstrates that the model was significantly fitted to the CCD; in addition, the *p*-value of the “Lack of Fit” is 0.8008 (*p* > 0.05), which means non-significant to the model of CCD. Therefore, the significance of “Model” and “Lack of Fit” indicated that the model was reliable and adequate. Moreover, the determination coefficient (R^2^) is 0.9932, and the adjusted R^2^ is 0.9872; that is, the difference between them is <0.2, which implies the model’s suitability to predict TPC, and fits to the current study [20]. The ANOVA results listed in Table 1 showed that the linear coefficient (X_1_, X_2_, X_3_), quadratic term coefficients (X_1_^2^, X_2_^2^, X_3_^2^), and interactive coefficient (X_1_ X _2_, X_1_ X_3_) were remarkably significant (*p* < 0.01). In contrast, the interactive coefficient X_2_ X_3_ was insignificant (*p* > 0.05).

Figure 2 shows the 3D response surfaces of TPC extracted from LSP varied with three individual parameters, visually illustrating the individual parameters’ interactive effects on the response.

Each plot of the interactive effects between two individual parameters on TPC is drawn while keeping the other variable at a fixed intermediate level. The interactive effect between the extraction time and liquid/solid ratio showed a remarkably negative effect on TPC (*p* < 0.0001) (Figure 1A, Table 1). The interactive effect between the extraction time and extraction temperature showed a significant positive effect on TPC (*p* < 0.01) (Figure 1B, Table 1). The interactive effect between the extraction temperature and liquid/solid ratio showed no significance on TPC (*p* > 0.05) (Figure 1C, Table 1). The TPC increased with the increase in extraction time and then showed a gentle or slightly decreasing trend, which was consistent with the results of single-factor experiments. A longer primary extraction time would expose the LSP to the extraction system, rupture the cell wall, and might release more intracellular components, while further extension of the extraction time might lead to the degradation of TPC [21,22]. The liquid/solid ratio played a positive role in TPC. The TPC increased with the initial increase in the liquid/solid ratio, while exceeding the specific range, and a decrease in TPC was displayed, especially with a prolonged extraction time. The present effect could be explained as follows: within the certain range, more water led to a higher relative contact area, and the interaction between water and phenolic compounds may be enhancive and thus increase the dissolution capacity of intracellular components [23]; exceeding the certain range, more water might increase the amount of oxygen dissolved in the extraction system, which could increase the oxidation of phenolic compounds at high temperatures with a long extraction time [24].

Extraction temperature also played an important role in TPC. Basically, at a certain threshold, a higher temperature almost increases TPC. This is due to the faster mass transfer in the extraction system and thus results in facilitating the phenolic compounds diffusion of LSP under thermal activation [25]. Still, the structural decomposition of phenolic compounds occurs at high temperatures, according to the reported studies, which led to a decrease in TPC. This phenomenon could be explained as follows: within the certain range, the extraction of phenolic compounds was only affected by thermal instability [26], while exceeding the certain range, the extraction of phenolic compounds was attributed to the oxidation and degradation [27], which led to the decrease in TPC.

#### 2.2.3. Model Validation

Based on the regression model, the optimal phenolic compounds extraction conditions of LSP using MCAE were an extraction time of 80 min, liquid/solid ratio of 42.8 mL/g, and extraction temperature of 80.7 °C. Under these optimized conditions, the theoretical maximum value of TPC was 106.37 mg GAE/g DW. The model validity was verified by measurements conducted in triplicate under optimal conditions with a slight modification, extraction time of 80 min, liquid/solid ratio of 43 mL/g, and extraction temperature of 81 °C, the experimental value of TPC was 106.15 ± 1.44 mg GAE/g DW. Obviously, there was no statistically significant difference between the experimental value and the model predicted value with an error coefficient of 0.21%. The result indicated that the model was reliable for the extraction process.

### 2.3. Analysis of Antioxidant Activities

Since most of the antioxidant activities in LSP are provided by their phenolic compounds, which have been associated with some health benefits [28], antioxidant activities could be an essential indicator to consider regarding LSP. Therefore, three in vitro antioxidant indexes were employed to evaluate the antioxidant activities of phenolic compounds of LSP, namely, DPPH, ABTS, and FRAP [29]. Actually, no in vitro index is appropriate for quantifying all antioxidants because each antioxidant assay presents a different reaction mechanism. For DPPH and ABTS, the mechanism is based on hydrogen atom transfer and single electron transfer [30]. FRAP’s mechanism is based on single electron transfer, which was used to measure the potential of the phenolic compounds chelating Fe^2+^ in LSP to prevent oxidative damage [31,32]. The results of DPPH and ABTS assay were 279.75 ± 18.71 µmol TE/g and 618.60 ± 2.70 µmol TE/g, respectively, which is lower than the previous work [33]. Regarding the FRAP assay, the value was 634.14 ± 7.17 µmol TE/g, exceeding results from previous studies. This phenomenon may be due to the correlation that FRAP was highly correlated with phenolic acids and flavonoids [34], and the composition of the extracted phenolic compounds under optimized conditions of MCAE was analyzed using UPLC-triple-TOF/MS.

A total of eight phenolic compounds were identified from the LSP extract: gallocatechin, protocatechuic acid, hyperoside, quercetin 3-*O*-glucuronide, narirutin, hesperidin, kaempferol 3-glucuronide, and isorhamnetin 3-*O*-glucuronide. These identifications (Table 2) were based on spectral data from literature data and MS database references [34,35,36]. Among these eight phenolic compounds, seven are flavonoids and one is phenolic acid, which could explain the FRAP assay results compared to our previous work. Figure 3 shows that the primary component was quercetin 3-*O*-glucuronide, which agrees with our previous work, indicating that MCAE is effective for extracting phenolic compounds from LSP.

### 2.4. In Vitro Enzyme Inhibitory Activities

Macroscopically, α-amylase plays a crucial role in starch hydrolysis, influencing blood sugar levels post-meal because of starch digestion. Meanwhile, the reducing sugars from starch digestion are hydrolyzed by α-glucosidase into glucose, further affecting glucose metabolism [37]. Therefore, regulating enzyme activity is advisable for preventing postprandial hyperglycemia [38,39]. Previous studies have shown that phenolic compounds can inhibit α-amylase activity, which results in retarded starch digestion [40]. In addition, phenolic compounds were also investigated for the potential inhibitory action towards α-glucosidase [41,42]. Due to the abundant phenolic compounds in LSP extract, it was used to evaluate the inhibitory activity of α-amylase and α-glucosidase. Four addition methods for LSP extract were investigated to optimize in vitro enzyme inhibitory. As shown in Figure 4, significant differences in inhibitory effects were observed among these methods: (I) mixing LSP extract with the enzyme before adding starch; (II) mixing LSP extract with starch before adding enzyme; (III) mixing starch with the enzyme before adding LSP extract; and (IV) mixing all three components simultaneously. The I-adding method showed the greatest inhibitory effect on both α-amylase and α-glucosidase. This phenomenon could be due to a longer interaction time, allowing more contact between the phenolic compounds and the enzyme [43]. The II and IV addition methods having the same interaction time, showed similar inhibitory effects. The inhibition mechanism of the II addition method might involve starch/polyphenol complexes formed between the phenolic compounds in the LSP extract and starch [44,45]. These findings suggest that the LSP extracts inhibit enzymes through interaction between phenolic compounds and enzymes and rely on the starch/polyphenol complexes. In view of the inhibitory effect, the I addition method was the most effective, even though the inhibitory rate of the II and IV addition methods was about 80%. Therefore, phenolic compounds from LSP can be considered as a potential functional food to treat diabetes by retarding the increase in blood glucose, which can reduce postprandial hyperglycemia [38,43].

### 2.5. Antibacterial Activities of LSP Extracts

Thus far, antibiotics and antimicrobial agents have been used to inhibit pathogenic bacteria. However, inappropriate and irrational use has led to the appearance of resistant strains, especially foodborne pathogens, such as *Bacillus cereus*, *S. aureus,* and *E. coli*, which posed a threat to public health [46,47].

Three bacterial strains (*S. aureus*, *B. subtilis,* and *E. coli*) were used to assess the antibacterial activities of LSP extracts. Among the three bacterial strains were two Gram-positive bacteria (*S. aureus* and *B. subtilis*) and one Gram-negative bacteria (*E. coli*). The results of the antibacterial activities obtained from agar disc diffusion assays are shown in Table 3 and Figure 5. The LSP extracts were active against all types of bacteria tested. The diameters of inhibition zones of LSP extracts to *E. coli*, *S. aureus,* and *B. subtilis* were 15 mm, 12 mm, and 19 mm, respectively. LSP extracts showed a higher sensitivity to *B. subtilis* than *E. coli* and *S. aureus*. The phenomenon could be due to the presence of lipopolysaccharides structurally in the cell membrane of the Gram-negative bacteria, which could be a credible barrier against antibacterial agents [48,49].

As we know, microorganisms impact on food safety during food storage. It was reported that phenolic compounds in plants, such as grape seeds and bagasse extract, have antimicrobial properties, which could be due to the degradation of cell walls, disruption of protein translocation, or altered phospholipid components [50]. Based on the above results of in vitro biological activities, LSP extracts could be used as a potential natural additive to extend the shelf life of food products.

## 3. Materials and Approaches

### 3.1. Materials and Chemical Reagents

The lotus seedpod was obtained from the lotus (*Nelumbium speciosum*) processing by-product purchased from Quzhou, Zhejiang province, China. The lotus seedpod was dried by air in a stove at 45 °C for 48 h, reaching a constant weight, and subsequently ground into powder by an XL-04B pulverizer (Guangzhou, China) and sieved by 40-mesh sieves. Before extraction, the powder was stored at −20 °C.

All the analytical grade reagents used in this study were purchased from the Aladdin Chemistry Company (Shanghai, China). All other analytical grade reagents, such as gallic acid, ABTS, CoCO_3_, BaCO_3_, and Li_2_CO_3_ were purchased from Macklin Chemistry Company (Shanghai, China).

### 3.2. MCAE

Eight solid reagents were evaluated in the MCAE process: BaCO_3_, Na_2_B_4_O_7_·10H_2_O, K_2_CO_3_, Li_2_CO_3_, NaHCO_3_, Na_2_CO_3_, CaCO_3_, and CoCO_3_. First, 150 g stored LSP powder was ground again by an AM400 planetary ball mill (Ants Scientific Instruments, Beijing, China); two drums and 500 mL each; 10 mm stainless steel ball; at 400 r/m for 10 min; second, ground LSP power 2 g was mixed with some extent content of solid reagents (25%, g/g) and co-grinding by the AM400 planetary ball mill with a 25 mL reaction chamber and 10 mm stainless steel ball for 10 min to obtain a fine powder. After co-grinding, 0.5 g mixtures were added to double-neck flat-bottomed flasks containing a water-cooling system and a magnetic stirring bar (TP-350+, MIULAB, Hangzhou, China), then stirred and extracted by distilled water at a specific temperature (range of 30 °C to 90 °C) for a while (range of 10 min to 90 min). After centrifuging at 5000 r/m for 10 min,

### 3.3. Yield of TPC (Y_TP_)

The Folin–Ciocalteu method was used to measure the Y_TP_ of LSP [51] with some modifications. In brief, 0.5 mL of diluted extract was mixed with 2.5 mL of 10-fold-diluted Folin–Ciocalteu reagent; after 5 min away from light, 2 mL of Na_2_CO_3_ solution (7.5%, *w*/*v*) was added. After being vortexed and incubated away from light for 60 min, the mixture was measured by a spectrophotometer (Shanghai Spectrum Instruments Co., Ltd., SP-723, Shanghai, China) at 765 nm, and the result was expressed as mg of gallic acid equivalent per g dry weight (mg GAE/g DW) of LSP.

### 3.4. Experimental Design

After screening through MCAE experiments described in Section 3.2, Li_2_CO_3_ was considered the most efficient solid reagent. Based on the selected Li_2_CO_3_, the influence of every single factor, namely, extraction time (10–90 min), liquid/solid ratio (20–60 *v*/*v*), and extraction temperature (30–90 °C), on the response value of total phenolic compounds was studied through single-factor experiments. The range of factors was determined according to the value of TPC close to the optimal region. The RSM was designed to optimize the experiment levels of variables. A central composite design with a three-factor (X_1_, extraction time; X_2_, liquid/solid ratio and X_3_, extraction temperature) (Table 4) was applied for the response of Y_TP_. Herein, a total of 20 experimental set-ups were performed and the second-order quadratic model was determined as follows:*Y* = k_0_ + k_1_X_1_ + k_2_X_2_ + k_3_X_3_ + k_11_X_1_X_2_ + k_12_X_1_X_3_ + k_13_X_2_X_3_ + k_11_X_1_^2^ + k_22_X_2_^2^ + k_33_X_3_^2^(2)
where *Y* is the predicted TPC; k_0_, k_1_–k_3_, k_11_–k_13_, and k_11_–k_33_ are the interception, linear, interacting, and quadratic coefficients, respectively.

### 3.5. UPLC-Triple-TOF/MS Analysis

Qualitative identification analysis was evaluated by UPLC-Triple-TOF/MS 6600plus system (AB SCIEX Co., Framingham, MA, USA) for the phenolic compounds in LSP extracts according to the previous protocol [52], with slight modification. Five-microliter samples were injected and separated using a ZORBAX SB-C_18_, 250 mm × 4.6 mm,1.8 µm column (Agilent Technologies Co., Santa Clara, CA, USA) with a constant flow rate of 0.65 mL/min, and the column temperature was maintained at 28 °C. The UV detector was set at 280 nm. The mobile phases consisted of solvent A (acetic acid: water = 1:99) and solvent B (acetonitrile), and the elution gradient system was as follows: 0–11 min, 8% B; 11–15 min, 9% B, 15–25 min, 15% B; 25–28 min, 20% B; 28–31 min, 40% B; 31–45 min, 60% B; 45–65 min, 8% B. The scan range of the MS condition was 100–1500 *m*/*z*. Peak View software (version 1.2, SCIEX, Framingham, MA, USA) was used to analyze the data, and the phenolic compounds were identified by using the fragment and molecular ion peaks, confirmed in previous literature.

### 3.6. In Vitro Antioxidant Activities

DPPH assay was used to determine the DPPH radical scavenging capacity of the samples according to the descriptions by Brand-Williams [53], with minor changes. An aliquot of 1 mL of extract was mixed with a 2 mL DPPH working solution (0.2 mmol/L ethanol). The reaction was carried out in the darkness at ambient temperature for 30 min, and the absorbance of each sample was measured at 517 nm (Spectrum Instruments, SP-723, Shanghai, China). The standard curve was drawn using Trolox (0–0.1 mmol/L), and the results were expressed as Trolox equivalent (TE) per gram dry weight of LSP (mg TE/g DW).

The method of the ABTS radical scavenging capacity was described in our previous work [52], with minor changes. A total of 10 mL of 2.0 mmol/L ABTS solution and 0.1 mL of 70 mmol/L potassium persulfate solution were mixed, vortexed, and incubated in darkness for 12–16 h at room temperature. Then, the ABTS working solution was obtained by diluting with absolute ethanol (the absorbance value was 0.7 ± 0.02 measured at 734 nm using a spectrophotometer). Subsequently, 0.15 mL of each sample was mixed with 1.5 mL ABTS working solution into a 2 mL tube incubated in darkness for 6 min after vortex, and the absorbance value was measured at 734 nm. The standard curve was drawn using Trolox (0–0.333 mmol/L), and the results were expressed as TE per gram dry weight of LSP (mg TE/g DW).

The method of FRAP assay was determined by Oldoni [54], with minor changes. In brief, the standard curve was drawn using Trolox (0–0.5 mmol/L), and the results were expressed as TE per gram dry weight of LSP (mg TE/g DW).

### 3.7. In Vitro Enzyme Inhibitory Activities

The method of α-amylase inhibitory activity was determined, as reported by Zhang [43]. In brief, 50 μL of LSP extracts [I], 5 mg/mL soluble starch solution [S], and 3 U/mL α-amylase aqueous solution [E] were mixed and incubated at 37 °C for 20 min. Subsequently, the remainder was added and incubated in tubes at 37 °C for 3 min; there are four different methods of adding samples. After inactivating enzymes in a water bath (90 °C) for 1 min, 100 μL 3,5-Dinitrosalicylic acid (DNS) reagent solution was subsequently added and incubated at 90 °C for 10 min. The mixture was diluted with 900 μL distilled water, and the results were determined at 540 nm (Thermo Fisher Scientific, 1510, Shanghai, China).

The method of α-glucosidase inhibitory activity was also determined following the method reported by Zhang [43]. Briefly, 20 μL of LSP extracts [I_1_], 2.5 mmol/L 4-*N*-trophenyl-α-d-glucopyranoside (pNPG) [S_1_], and 3.38 U/mL α-glucosidase [E_1_] were mixed and incubated at 37 °C for 20 min. Then, the mixture was added to 120 μL phosphate buffer solution (PBS) (0.1 mol/L, pH 6.8) and incubated at 37 °C for 10 min, i.e., there are four different ways of adding samples (same as above). After terminating the reaction by adding 80 μL of 0.2 mmol/L sodium carbonate solution, the results were measured at 405 nm on the Thermo Fisher Scientific.

### 3.8. In Vitro Antibacterial Activities

Three foodborne bacterial pathogens including Gram-positive bacteria (*Staphylococcus aureus* and *Bacillus subtilis*) and Gram-negative bacteria (*Escherichia coli*) were used to assess the antibacterial activities of LSP extracts.

Luria–Bertani (LB) nutrient broth was used to grow all three bacteria mentioned above. After incubating at 37 ± 0.1 °C for 20 h, the concentration of bacterial suspension was adjusted to about 1.0 × 10^7^ colony forming units (CFU/mL). A total of 10 mL of bacterial suspension was mixed with 150 mL nutrient agar (sterilized in a conical flask and cooled to roughly 50 °C). The mixture was swirled (to distribute the medium homogeneously) and inoculated to sterilized the Petri dishes (9 cm-diameter, approximately 20 mL/Petri dish). After cooling, four symmetrical holes were drilled for each Petri dish, and approximately 150 μL of LSP extract (diluted 0, 2, 4, and 8 times, respectively) was injected into the holes. Subsequently, the treated Petri dishes were incubated at 37 ± 0.1 °C for 16–20 h, and the inhibition zones, measured by slide caliper, were applied to assess the in vitro antibacterial activities of the LSP extracts.

### 3.9. Statistical Analysis

All the experiments were performed at least three times, and the results were expressed as means ± standard deviation (SD) (*p* < 0.05 was defined as statistically significant, which was determined by one-way ANOVA with Tukey’s test); IBM SPSS statistics 20 was used to conduct the statistical analysis. Design Expert software version 8.0.6 (Stat-Ease Inc., Minneapolis, MN, USA) was used for the RSM experimental design, regression coefficient analysis and graphical analysis. GraphPad Prism 8 software (GraphPad Software Inc., San Diego, CA, USA) was performed for other statistical plots.

## 4. Conclusions

The present study developed a “green and eco-friendly” method for phenolic compounds extraction from LSP using water as an extraction medium. The optimal MCAE conditions were identified as 80 min, 42.8 mL/g, and 80.7 °C, with Li_2_CO_3_ as the most effective solid regent. Under these optimized conditions, eight phenolic compounds were identified using UPLC-Triple-TOF/MS. Moreover, the results demonstrated that the phenolic compounds extracted from LSP using the MCAE method possessed significant biological activities, as evidenced by in vitro antioxidant, enzyme inhibitory, and antibacterial activities. Given their biological activities, the phenolic compounds could potentially serve as natural additives to extend the shelf life of food products. This work is fundamental research for recovering phenolic compounds from LSP using the MCAE method and evaluating their biological activities. It provides theoretical insights for further research and development of lotus seedpod resources, potentially aiding environmental efforts to mitigate waste degradation.

## Figures and Tables

**Figure 1 molecules-28-07947-f001:**
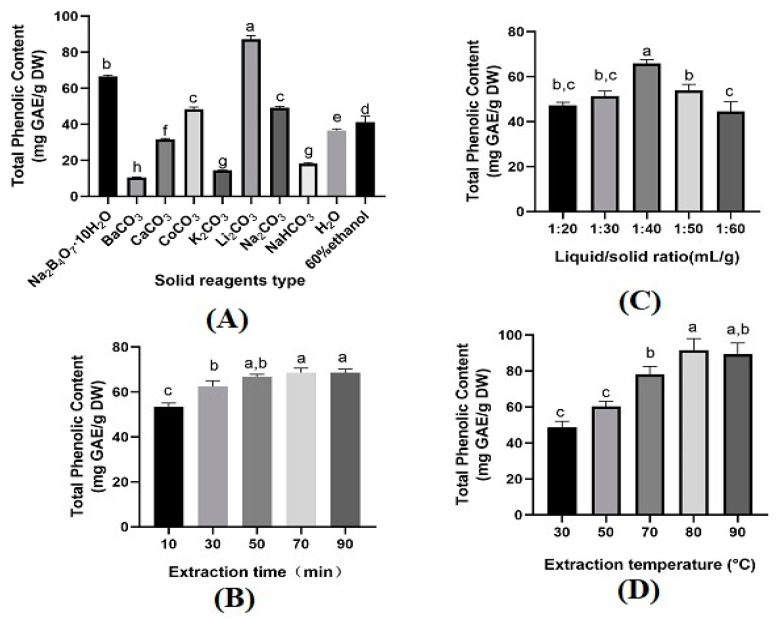
Effect of different solid reagents type (**A**), extraction time (**B**), liquid/solid ratio (**C**), and extraction temperature (**D**) on TPC. Data are presented as means ± SD (*n* = 3). The letters (a–h) above the bar show a significant difference at *p* < 0.05.

**Figure 2 molecules-28-07947-f002:**
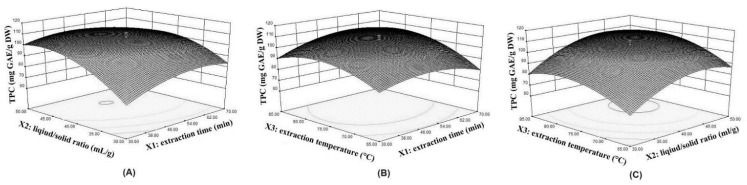
Response 3D surface plots on total phenolic content. (**A**) extraction time and liquid/solid ratio; (**B**) extraction time and; and (**C**) liquid/solid ratio and extraction temperature.

**Figure 3 molecules-28-07947-f003:**
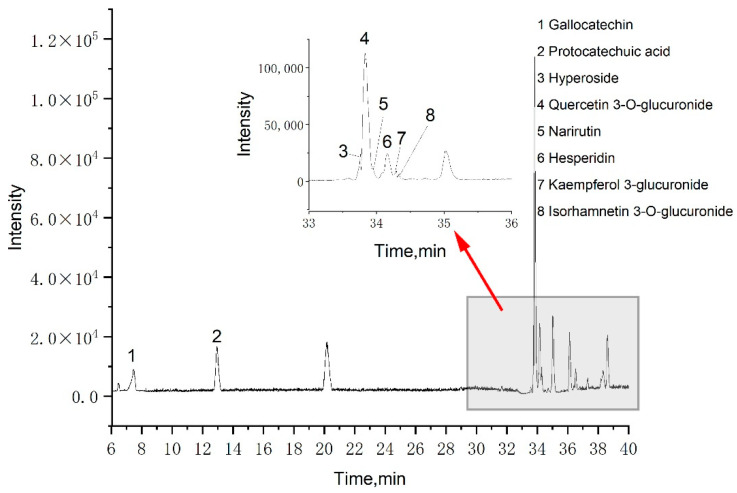
Basic peak chromatograms of LSP extract obtained by the MCAE method.

**Figure 4 molecules-28-07947-f004:**
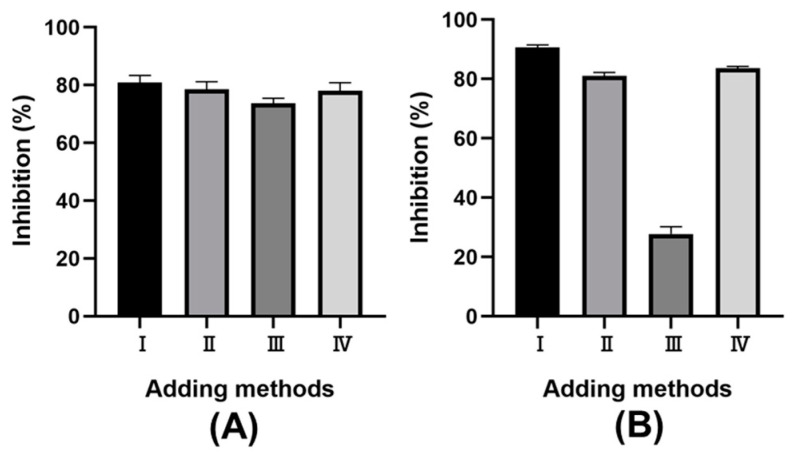
The effect of sample adding methods on the rate of inhibition. (**A**) α-amylase; (**B**) α-glucosidase.

**Figure 5 molecules-28-07947-f005:**
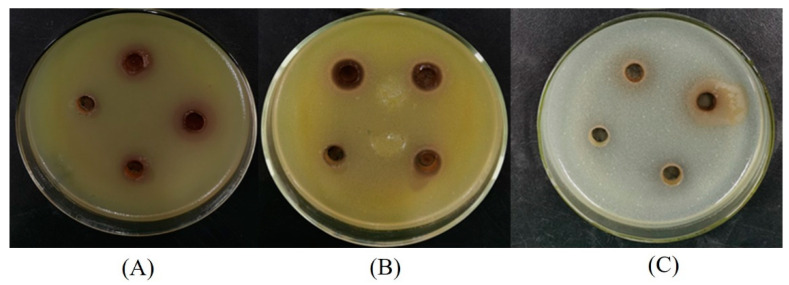
Diameters of inhibition zones of the LSP extract against the bacteria. (**A**) *E. coli*, (**B**) *S. aureus*, and (**C**) *B. subtilis*.

**Table 1 molecules-28-07947-t001:** Analysis of variance (ANOVA) for the CCD quadratic polynomial regression model.

Term	Sum of Squares	Df	Mean Square	F-Value	*p*-Value
Model	6766.85	9	751.87	163.3	<0.0001
Linear					
X_1_-extraction time	93.13	1	93.13	20.23	0.0011
X_2_-liquid/solid ratio	1645.33	1	1645.33	357.35	<0.0001
X_3_-extraction temperature	858.17	1	858.17	186.39	<0.0001
Interactions					
X_1_ X_2_	179.96	1	179.96	39.09	<0.0001
X_1_ X_3_	110.39	1	110.39	23.98	0.0006
X_2_ X_3_	6.66	1	6.66	1.45	0.2567
Quadratic					
X_1_^2^	918.02	1	918.02	199.39	<0.0001
X_2_^2^	2087.57	1	2087.57	453.4	<0.0001
X_3_^2^	1596.78	1	1596.78	346.81	<0.0001
Residual	46.04	10	4.60		
Lack of Fit	14.24	5	2.85	0.45	0.8008
Pure Error	31.81	5	6.36		
Cor Total	6812.90	19			
Std. Dev.	2.15				
R^2^	0.9932				
Adjusted R^2^	0.9872				
Adeq Precision	34.603				
Mean	86.72				
C.V.%	2.47				

**Table 2 molecules-28-07947-t002:** Identification of phenolics in optimized MCAE condition by UPLC-Triple-TOF/MS.

Peak	Rt (min)	[M − H]^−^ (*m*/*z*)	MS^2^ Ions (*m*/*z*)	Compound
1	7.455	305.0663	125.0234, 179.0335	gallocatechin
2	12.970	153.0194	108.0216, 109.0287	protocatechuic acid
3	33.752	463.0872	300.0274, 271.0244, 255.0292, 301.0345	hyperoside
4	33.835	477.0673	301.0360, 151.0037	quercetin 3-*O*-glucuronide
5	33.947	579.1712	271.0612, 151.0042	narirutin
6	34.162	609.1826	301.0737, 285.0369	hesperidin
7	34.286	461.0710	285.0400, 229.0505	kaempferol 3-glucuronide
8	34.339	491.0815	315.0513, 300.0275	isorhamnetin 3-*O*-glucuronide

**Table 3 molecules-28-07947-t003:** Antibacterial activity of LSP extracts against the bacteria.

Type	*E. coli*	*S. aureus*	*B. subtilis*
Inhibition zone (mm)	15	12	19

**Table 4 molecules-28-07947-t004:** The central composite design and values of total phenolic content extracted from LSP.

	Independent Variables				
Run Order	Extraction Time (min)	Liquid/Solid Ratio (mL/g)	Extraction Temperature (°C)	Experimental Values of TPC (mg GAE/g DW)	Predicted Values of TPC (mg GAE/g DW)
1	50	40	75	109.78	107.58
2	70	30	85	83.57	84.14
3	70	50	65	72.70	73.33
4	83.6	40	75	90.04	89.40
5	50	40	75	105.67	107.58
6	16.4	40	75	82.22	80.61
7	50	23.2	75	56.51	55.08
8	50	40	58.2	65.97	64.48
9	30	50	85	94.55	95.27
10	50	40	75	107.72	107.58
11	50	40	75	105.67	107.58
12	30	50	65	84.01	85.02
13	50	40	75	111.22	107.58
14	50	40	91.8	91.89	91.14
15	50	40	75	105.05	107.58
16	70	30	65	61.82	62.69
17	50	56.8	75	92.81	92.00
18	30	30	65	54.11	55.41
19	70	50	85	98.15	98.44
20	30	30	85	61.05	62.01

## Data Availability

All data generated or analyzed during this study are included in this published article.

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
