# Peer review of "Mechanochemical-Assisted Extraction and Biological Activity Research of Phenolic Compounds from Lotus Seedpod (Receptaculum Nelumbinis)"

_molecules, 2023, doi:10.3390/molecules28247947_

Round 1
Reviewer 1 Report
Comments and Suggestions for Authors
This article developed an efficient mechanochemical-assisted extraction process and assessed the bioactivities of polyphenols from lotus seedpod. The current study is on a topic of relevance and general interest to the journal's readers. However, the descriptions of some very important points were inadequate or completely missing, leaving some confusion for readers. Here are my specific suggestions.
1. The abstract clearly outlines the research's purpose, methods, and main findings. However, it could be more concise and focused on key results. Consider shortening it to enhance readability.
2. Does the maturity and drying parameters of lotus seed pod affect the content of active substances in the raw materials? How was it determined in this experiment. In addition, the choice of Li2CO3 as the most efficient solid reagent is mentioned, but a more detailed explanation of why this was the case would strengthen the methodology section. It might be useful to include comparative data showing the efficacy of different reagents
3. In the discussion section, more comparative analysis with existing literature would strengthen the argument about the efficiency and novelty of your extraction method.
4. Given the focus on a green extraction process, a section discussing the environmental impact or benefits of using MCAE compared to traditional methods would be helpful.
5. Discuss the potential applications of your findings in industrial or practical settings. This could include potential uses of the extracted polyphenols or the scalability of the extraction method.
Comments on the Quality of English Language
Line 42, should be "plant materials"
Line 52, should be " [12], and"
Line 62, in vitro should be italics
Line 75,should be One-hundred and fifty grams of ..
Line 76 and 81, for r/m and rpm, please unify it throughout manuscript
Generally, there are lots of grammatical and format errors typos scattered throughout the manuscript. Please check and revise them.
Author Response
Point 1: The abstract clearly outlines the research's purpose, methods, and main findings. However, it could be more concise and focused on key results. Consider shortening it to enhance readability. 

Response 1: Thanks for your suggestions. We have revised the abstract again. We have changed the word “polyphenols” into “phenolics”, and changes were made throughout the text wherever necessary. We have deleted the sentence “Subsequently, the identification of polyphenols was carried out using triple-time-of-flight mass spectrophotometry (UPLC-Triple-TOF/MS). Besides, the biological activities of polyphenols were evaluated through in vitro antioxidant, enzyme inhibitory, and antibacterial activities.” We have changed “106.37 mg GAE/g DW” into “106.15±1.44 gallic acid equivalents (GAE)/g dry weight (DW)”. (Line19) We have changed the “the (2,2-Diphenyl-1-picrylhydrazyl) DPPH, 2,20-azinobis (3-ethylbenzothiazoline-6-sulfonic acid) (ABTS)”into “2,2-Diphenyl-1-picrylhydrazyl (DPPH), 2,2’-azinobis (3-ethylbenzothiazoline-6-sulfonic acid) (ABTS) ” (Line20-21). We have now added the sentence “Triple-time-of-flight mass spectrophotometry (UPLC-Triple-TOF/MS) analysis identified 8 phenolics that mainly consisted of polyphenols and flavonoids.” (Line22-23). We have changed the word “strong” into “potent”. (Line 24). All the changes are modified and highlighted with red colour by using the "Track Changes" function in Microsoft Word.
Point 2: Does the maturity and drying parameters of lotus seed pod affect the content of active substances in the raw materials? How was it determined in this experiment. In addition, the choice of Li2CO3 as the most efficient solid reagent is mentioned, but a more detailed explanation of why this was the case would strengthen the methodology section. It might be useful to include comparative data showing the efficacy of different reagents.
Response 2: Thanks for your suggestions. Yes, the maturity and drying parameters of lotus seed pod affect the content of active substances in the raw materials. So, we have added the sentence “reaching a constant weight” (Line 1360). We have changed “Compared with conventional extraction by using (water and 60% ethanol), the values of TPC obtained by MCAE (Na2B4O7·10H2O, Li2CO3, CaCO3 and CoCO3) were higher, and (NaHCO3, K2CO3, and BaCO3) were lower. As for the tested solid reagents, Li2CO3 was found to be the most effective one for the extraction of TPC (87.24 ±1.98 mg GAE/g DW), which was significantly higher than Na2B4O7·10H2O (66.65±0.73 mg GAE/g DW) or Na2CO3 (49.30±0.63 mg GAE/g DW (Figure 1(A)). It might imply that polyphenols salts were easier formed under Li2CO3 than Na2B4O7·10H2O or Na2CO3 mechanochemical pretreatment, or might due to when extracted polyphenols with Li2CO3, the polyphenols did not exhibit antioxidant power. [10]” into “To evaluate the role of MCAE in extraction process, conventional extraction using 60 % ethanol was also investigated under the same extraction conditions. Compared with water or 60% ethanol, the values of TPC obtained by MCAE (Na2B4O7·10H2O, Li2CO3, CaCO3 and CoCO3) were higher, especially Na2B4O7·10H2O and Li2CO3, which were remarkably higher, but NaHCO3, K2CO3, and BaCO3 were not ideal which were lower than them. The value of TPC obtained by MCAE-Li2CO3 was 87.24 ± 1.98 mg GAE/g DW, which was significantly higher than Na2B4O7·10H2O (66.65±0.73 mg GAE/g DW) or Na2CO3 (49.30±0.63 mg GAE/g DW (Figure 1(A)), not to mention water (36.63 ± 0.82 mg GAE/g DW) or 60 % ethanol (41.26 ± 3.29 mg GAE/g DW). This phenomenon may be summarized into two reasons: (1) Li2CO3 can destroy or disintegrate the LSP cells wall easier [15,16]; (2) phenolics salts may be easier and more accessible formed under Li2CO3 than Na2B4O7·10H2O or Na2CO3 mechanochemical pretreatment, which can improve the water-solubility of phenolics [15,17]. ”(Line 371-Line383)
Point 3: In the discussion section, more comparative analysis with existing literature would strengthen the argument about the efficiency and novelty of your extraction method.
Response 3: Thanks for your suggestions. We have now added “As so far, MCAE was widely used in extraction bioactive components, including phenolics, from plant materials. A study reported that phenolic compounds was extracted from Laurus nobilis by using MCAE- Li2CO3 compared with conventional methods [10], the value of phenolic compounds by these two methods is similar. However, extraction time of MCAE- Li2CO3 is reduced more than ten times compared with conventional method. The study by Xu et al reported their work on reducing energy and solvent consumption, moreover, might realize the complete utilization of the plant material, which was proved to be an environmentally friendly and effective extraction method [18].” (Line383-Line391)
“
Point 4: Given the focus on a green extraction process, a section discussing the environmental impact or benefits of using MCAE compared to traditional methods would be helpful.
Response 4: Thanks for your suggestions. We have now added “As so far, MCAE was widely used in extraction bioactive components, including phenolics, from plant materials. A study reported that phenolic compounds was extracted from Laurus nobilis by using MCAE- Li2CO3 compared with conventional methods [10], the value of phenolic compounds by these two methods is similar. However, extraction time of MCAE- Li2CO3 is reduced more than ten times compared with conventional method. The study by Xu et al reported their work on reducing energy and solvent consumption, moreover, might realize the complete utilization of the plant material, which was proved to be an environmentally friendly and effective extraction method [18].” (Line383-Line391)
Point 5: Discuss the potential applications of your findings in industrial or practical settings. This could include potential uses of the extracted polyphenols or the scalability of the extraction method.
Response 5: Thanks for your suggestions. We have now added “As so far, MCAE was widely used in extraction bioactive components, including phenolics, from plant materials. A study reported that phenolic compounds was extracted from Laurus nobilis by using MCAE- Li2CO3 compared with conventional methods [10], the value of phenolic compounds by these two methods is similar. However, extraction time of MCAE- Li2CO3 is reduced more than ten times compared with conventional method. The study by Xu et al reported their work on reducing energy and solvent consumption, moreover, might realize the complete utilization of the plant material, which was proved to be an environmentally friendly and effective extraction method [18].” (Line383-Line391)
Point 6: Comments on the Quality of English Language
Line 42, should be "plant materials"
Line 52, should be " [12], and"
Line 62, in vitro should be italics
Line 75,should be One-hundred and fifty grams of .
Line 76 and 81, for r/m and rpm, please unify it throughout manuscript
Response 6: Thanks for your suggestions. We have now changed "plant material" into "plant materials" in Line 42; added “,” before and in Line 52; changed “in vitro” into “in vitro”, and changes were made throughout the text wherever necessary; changed 150 g into “One-hundred and fifty grams”; changed “rpm” into “r/m” and unified it throughout manuscript.
Point 7: Generally, there are lots of grammatical and format errors typos scattered throughout the manuscript. Please check and revise them.
Response 7: Thanks for your suggestions. We have now revised this manuscript written in standard by native speaker. This manuscript is modified more than 160 places and highlighted with red colour by using the "Track Changes" function in Microsoft Word.

Reviewer 2 Report
Comments and Suggestions for Authors
Title: the title should be reformulated because the obtained extracts do not contain only polyphenols and the authors cannot claim that the shown activities are related to only polyphenols.
Abstract and keywords: The full Latin name of the plant should be given in the brackets after the common name; line 16, in vitro should be in italics; line 18, between number and unit min should be space; the authors should give a real (measured) value of the TPC using the mentioned conditions; the full meaning of GAE and DW should be given; lines 20-21, check typing mistakes, brackets, long name or abbreviation in the brackets?; line 22, between number and unit should be space; line 22, the word ''assay'' should be deleted; line 25, the full stop should be deleted; line 26, the author should use B. subtilis instead of the full name; it should not be the abbreviation in the keywords.
Introduction: line 33, ‘’Gaertn.’’ should not be in italics; the numbers of the references should be before the full stops, not after; line 36, instead of ‘’and so on’’ should be given more examples; lines 37 and 38, the usage of the word ‘’activity’’ should be reduced, thus the sentence should be reformulated; line 51, flavonoids are also antioxidant phenolic compounds; line 52, the abbreviation should be introduced in line 149 after full meaning; line 58, ‘’the bioactivities of polyphenols’’ should be reformulated because the obtained extracts do not contain only polyphenols and the authors cannot claim that the showed activities are related to only polyphenols; instead of solid solvent should be solid regent; line 62, in vitro should be in italics in the whole manuscript; lines 62-63, the usage of the word ‘’activities’’ should be reduced, thus the sentence should be reformulated; the used plant should be explained in more detail - botanical characteristics, habitat, chemical composition, plant family, potential products on the market, etc.; the full meaning of RSM should be given; the authors should point out the novelty of their research because it seems that there is a lack of novelty.
Material and Methods: this section should be after Results and Discussion (according to the journal rules); between the unit °C and number should not be space, but between number and other units should be space, line 68 and in the rest of the text; line 72, ‘’and’’ is missed; the sentence should not star using the number; line 76, the country is missed; line 77, which solid reagents? The preparation and types of solid reagents used in the extraction procedure should be explained in more detail; lines 80 and 81, ranges of the temperatures and extraction times should be given; line 81, the device is missed (name, producer, city, and country); line 92, ‘’After screening through MCAE experiments’’, the preliminary screening should be explained in more detail; also, ‘’Li2CO3 was considered as the most efficient solid reagent’’, but other solid reagents were not mentioned; line 93, ‘’Based on the above result’’, which results?; line 93, the authors should mention all single factors, as well as their ranges; line 102, ‘’and’’ is missed; the full meaning of TPC should be given in the title of Table 1; the units of all independent variables, as well as TPC should be given in Table 1; did the authors perform preparation of the extracts and determination of TPC in triplicate? Why does Table 1 not contain standard deviations after the values? Why did the authors explain the FC method and DPPH/ABTS antioxidant assays that were commonly used but did not explain UPLC-triple-TOF/MS analysis in more detail?; lines 133 and 134, [I] and [E] should be explained (as well as signs in lines 141-142); between number and unit should be space; line 136, it is not clear which four different methods of adding samples are used (the same for line 144); the full meaning of DNS, pNPG, and PBS should be given; the sentence should not start using the number; in vitro should be in italics; the authors did not explain the purification and elimination of the salts from the extracts; the extracts containing these salts cannot be used in food; the authors should mentioned all positive and negative controls using in all assays for the biological activities.
Results and Discussion: Mini introduction is missed in section 3.1.; Figure 1 - the full meaning of TPC is missed, as well as used statistical tool(s); the authors should explained why they used the mentioned levels of conditions (amount of solid reagent 25%, extraction time 30 min, liquid/solid ratio 20 mL/g, extraction temperature 30 °C, etc.); lines 183-185, check typing mistakes; lines 190-191 should be reformulated because salts do not influence polyphenol antioxidant capacity. Salts can significantly influence their release and concentration in the extraction medium but their antioxidant capacity depends on their structure. Additionally, Li2CO3 was the best but polyphenols did not exhibit antioxidant power. It is completely wrong and should be reformulated.; line 194, ''The influence factors'' should be changed in ''the influence of different factors''; it is not clear how CCD suggested 75°C that was not used but in the case of s-to-s ratio and time, it suggested the used values; discussion is completely missed in section 3.2.1.; the Eq.2. should be with bigger letters and numbers, and without + at the beginning; Table 2 - the full meaning of CCD should be given; Figure 2 - the full meaning of TPC should be given and the resolution should be significantly improved, the words are completely illegible; ''TPC, especially the polyphenols'' ??? but TPC is total polyphenol content; ''TPC extraction yield'' it should be TPC or extraction yield; line 245, ''the'' should be deleted; line 246, ''produced'' should be deleted; the English language should be significantly improved in the whole manuscript and checked by the native speaker; ''That is the scavenging activities of DPPH radical, ABTS radical and the FRAP.'' should be reformulated; the sentence ''It is well known that the DPPH analysis is evaluated by the DPPH radical scavenging potential of the polyphenols from LSP [30], and the ABTS assay is also based on the capture of free radicals, which is similarly to DPPH.'' should be deleted; the values of antioxidant potential obtained in other studies should be deleted; the presence of this sentence is not clear and the given explanation is not logical: ''The phenomenon of them may be attributed to the higher recovery of TPC, which was similar to the previous work.'' The presentation and discussion of the results of the antioxidant assays should be rewritten because, in this way, it is completely useless and not clear; the names of polyphenols should not be in uppercase letters; it is not clear which four adding methods for LSP extract were investigated; the discussion and comparison to the literature data are completely missed in section 3.5.
Conclusion: used extraction method cannot be named green;
References: check typing mistakes (uppercase letters, italics, etc.).
Comments on the Quality of English LanguageThe English language should be significantly improved in the whole manuscript and checked by the native speaker.
Author Response
Point 1: Title: the title should be reformulated because the obtained extracts do not contain only polyphenols and the authors cannot claim that the shown activities are related to only.

Response 1: Thanks for your suggestions. We have now reformulated the title by changing “Polyphenols” into “Phenolics”, and changes were made throughout the text wherever necessary.
Point 2: Abstract and keywords: The full Latin name of the plant should be given in the brackets after the common name; line 16, in vitro should be in italics; line 18, between number and unit min should be space; the authors should give a real (measured) value of the TPC using the mentioned conditions; the full meaning of GAE and DW should be given; lines 20-21, check typing mistakes, brackets, long name or abbreviation in the brackets?; line 22, between number and unit should be space; line 22, the word ''assay'' should be deleted; line 25, the full stop should be deleted; line 26, the author should use B. subtilis instead of the full name; it should not be the abbreviation in the keywords.
Response 2: Thanks for your suggestions. We have now added the Latin name of lotus seedpod in the brackets after the common name (Receptaculum Nelumbinis);We have changed “in vitro” into “in vitro”, and changes were made throughout the text wherever necessary; We have now added a space between number and unit, and changes were made throughout the text wherever necessary; We have now added the full meaning of gallic acid equivalents (GAE)and dry weight (DW); We have checked the typing mistakes and changed “(2,2-Diphenyl-1-picrylhydrazyl) DPPH, 2,20-azinobis 3-ethylbenzothiazoline-6-sulfonic acid) (ABTS)” into “2, 2-Diphenyl-1- picrylhydrazyl(DPPH), 2,2’-azinobis(3-ethylbenzothiazoline-6-sulfonic acid) (ABTS) ” ; We have deleted the word assay and the full stop; We have used B. subtilis instead of the full name; We have changed the MCAE into “mechanochemical-assisted extraction; phenolics” in the keywords.
Point 3: Introduction: line 33, ‘’Gaertn.’’ should not be in italics; the numbers of the references should be before the full stops, not after; line 36, instead of ‘’and so on’’ should be given more examples; lines 37 and 38, the usage of the word ‘’activity’’ should be reduced, thus the sentence should be reformulated; line 51, flavonoids are also antioxidant phenolic compounds; line 52, the abbreviation should be introduced in line 149 after full meaning; line 58, ‘’the bioactivities of polyphenols’’ should be reformulated because the obtained extracts do not contain only polyphenols and the authors cannot claim that the showed activities are related to only polyphenols; instead of solid solvent should be solid regent; line 62, in vitro should be in italics in the whole manuscript; lines 62-63, the usage of the word ‘’activities’’ should be reduced, thus the sentence should be reformulated; the used plant should be explained in more detail - botanical characteristics, habitat, chemical composition, plant family, potential products on the market, etc.; the full meaning of RSM should be given; the authors should point out the novelty of their research because it seems that there is a lack of novelty.
Response 3: Thanks for your suggestions. We have now changed “Gaertn.” into “Gaertn.”. We have changed the numbers of the references before the full stops, and changes were made throughout the text wherever necessary; We have reduced the usage of the word “activity” and the sentence has been reformulated as “Moreover, researchers have demonstrated that lotus seedpod has the activity of anti-oxidant [4], anti-inflammatory [5], anti-microorganisms [6], antiproliferative [7], and anti-gout [2], which is mainly because of rich in phenolics [4] and could lead it to be a new source of phenolics [8].” We have now changed “phenolic compounds” into “polyphenols”; We have now changed “solid solvent” into “solid regent”, and changes were made throughout the text wherever necessary; We have now reduced the usage of the word “activities” and the sentence has been reformulated as “Furthermore, in vitro antioxidant, enzyme inhibitory, and antibacterial activities of phenolics under optimized conditions were determined.” We have now added the Latin name of the plant as “Nelumbium speciosum”. We have described the novelty of our research, that is using MCAE method to extract phenolics from LSP and the biological activities were investigated.
Point 4: Material and Methods: this section should be after Results and Discussion (according to the journal rules); between the unit °C and number should not be space, but between number and other units should be space, line 68 and in the rest of the text; line 72, ‘’and’’ is missed; the sentence should not star using the number; line 76, the country is missed; line 77, which solid reagents? The preparation and types of solid reagents used in the extraction procedure should be explained in more detail; lines 80 and 81, ranges of the temperatures and extraction times should be given; line 81, the device is missed (name, producer, city, and country); line 92, ‘’After screening through MCAE experiments’’, the preliminary screening should be explained in more detail; also, ‘’Li2CO3 was considered as the most efficient solid reagent’’, but other solid reagents were not mentioned; line 93, ‘’Based on the above result’’, which results?; line 93, the authors should mention all single factors, as well as their ranges; line 102, ‘’and’’ is missed; the full meaning of TPC should be given in the title of Table 1; the units of all independent variables, as well as TPC should be given in Table 1; did the authors perform preparation of the extracts and determination of TPC in triplicate? Why does Table 1 not contain standard deviations after the values? Why did the authors explain the FC method and DPPH/ABTS antioxidant assays that were commonly used but did not explain UPLC-triple-TOF/MS analysis in more detail?; lines 133 and 134, [I] and [E] should be explained (as well as signs in lines 141-142); between number and unit should be space; line 136, it is not clear which four different methods of adding samples are used (the same for line 144); the full meaning of DNS, pNPG, and PBS should be given; the sentence should not start using the number; in vitro should be in italics; the authors did not explain the purification and elimination of the salts from the extracts; the extracts containing these salts cannot be used in food; the authors should mentioned all positive and negative controls using in all assays for the biological activities.
Response 4: Thanks for your suggestions. We have now put “Material and Methods” after “Results and Discussion”; We have deleted the space between unit °C and number, and changes were made throughout the text wherever necessary; We have now added “and” in the right place and changes were made throughout the text wherever necessary; the full meaning of TPC has been given in Abstract and we have now given the units of all independent variables; we had performed preparation of the extracts and determination of TPC in triplicate, but it had no occasion to do in triplicate for CCD; We have now added more details to explain UPLC-triple-TOF/MS analysis, and now added the sentence of (the fourth method was mixed the three together) to make the four different methods of adding samples more clear; We have now added the full meaning of 3,5-Dinitrosalicylic acid (DNS), 4-N-trophenyl-α-D-glucopyranoside (pNPG), and phosphate buffer solution (PBS). We have now changed 150 g into “One-hundred and fifty grams” and changed 10Ml into “Ten milliliters”. We do not explain the purification and elimination of the salts from the extracts, because this paper does not used LSP extracts in food.
Point 5: Results and Discussion: Mini introduction is missed in section 3.1.; Figure 1 - the full meaning of TPC is missed, as well as used statistical tool(s); the authors should explained why they used the mentioned levels of conditions (amount of solid reagent 25%, extraction time 30 min, liquid/solid ratio 20 mL/g, extraction temperature 30 °C, etc.); lines 183-185, check typing mistakes; lines 190-191 should be reformulated because salts do not influence polyphenol antioxidant capacity. Salts can significantly influence their release and concentration in the extraction medium but their antioxidant capacity depends on their structure. Additionally, Li2CO3 was the best but polyphenols did not exhibit antioxidant power. It is completely wrong and should be reformulated.; line 194, ''The influence factors'' should be changed in ''the influence of different factors''; it is not clear how CCD suggested 75°C that was not used but in the case of s-to-s ratio and time, it suggested the used values; discussion is completely missed in section 3.2.1.; the Eq.2. should be with bigger letters and numbers, and without + at the beginning; Table 2 - the full meaning of CCD should be given; Figure 2 - the full meaning of TPC should be given and the resolution should be significantly improved, the words are completely illegible; ''TPC, especially the polyphenols'' ??? but TPC is total polyphenol content; ''TPC extraction yield'' it should be TPC or extraction yield; line 245, ''the'' should be deleted; line 246, ''produced'' should be deleted; the English language should be significantly improved in the whole manuscript and checked by the native speaker; ''That is the scavenging activities of DPPH radical, ABTS radical and the FRAP.'' should be reformulated; the sentence ''It is well known that the DPPH analysis is evaluated by the DPPH radical scavenging potential of the polyphenols from LSP [30], and the ABTS assay is also based on the capture of free radicals, which is similarly to DPPH.'' should be deleted; the values of antioxidant potential obtained in other studies should be deleted; the presence of this sentence is not clear and the given explanation is not logical: ''The phenomenon of them may be attributed to the higher recovery of TPC, which was similar to the previous work.'' The presentation and discussion of the results of the antioxidant assays should be rewritten because, in this way, it is completely useless and not clear; the names of polyphenols should not be in uppercase letters; it is not clear which four adding methods for LSP extract were investigated; the discussion and comparison to the literature data are completely missed in section 3.5.
Response 5: Thanks for your suggestions. We have now added more discussion in section 3.1; We used the mentioned levels of conditions (amount of solid reagent 25%, extraction time 30 min, liquid/solid ratio 20 mL/g, extraction temperature 30 °C, etc.), because of the effective and enforceable. We have now reformulated the section 3.1 as “This phenomenon may be summarized into two reasons: (1) Li2CO3 can destroy or dis-integrate the LSP cells wall easier [15,16]; (2) phenolics salts may be easier and more accessible formed under Li2CO3 than Na2B4O7·10H2O or Na2CO3 mechanochemical pre-treatment, which can improve the water-solubility of phenolics [15,17]. As so far, MCAE was widely used in extraction bioactive components, including phenolics, from plant materials. A study reported that phenolic compounds was extracted from Laurus nobilis by using MCAE-Li2CO3 compared with conventional methods [10], the value of phenolic compounds by these two methods is similar. However, extraction time of MCAE-Li2CO3 is reduced more than ten times compared with conventional method. The study by Xu et al reported their work on reducing energy and solvent consumption, moreover, might realize the complete utilization of the plant material, which was proved to be an environmentally friendly and effective extraction method [18].” We have now changed ''The influence factors'' into ''the influence of different factors''; We had used 75°C in CCD because of the scope of CCD, and the temperature was set at 75°C that can cover more effective extraction temperature, so we closed 75°C for CCD. We have now changed Eq.2. with bigger letters and numbers, and deleted + at the beginning; We have now changed Figure 2 with higher resolution, and changed ''TPC extraction yield'' into “TPC”,and changes were made throughout the text wherever necessary; We have now deleted “the”, “produced” and the sentence of ''It is well known that the DPPH analysis is evaluated by the DPPH radical scavenging potential of the polyphenols from LSP [30], and the ABTS assay is also based on the capture of free radicals, which is similarly to DPPH.'' We have now reformulated the sentence ''That is the scavenging activities of DPPH radical, ABTS radical and the FRAP.'' as “namely, DPPH , ABTS,and FRAP [28]”; We have now revised this manuscript written in standard by native speaker. This manuscript is modified more than 160 places and highlighted with red colour by using the "Track Changes" function in Microsoft Word. We have now deleted the values of antioxidant potential obtained in other studies and rewritten the presentation and discussion of the results of the antioxidant assays to make it clearer. We have now changed the names of polyphenols in lowercase letters; We have now added the sentence of “The phenomenon might be due to the presence of structural lipopolysaccharides in Gram-negative bacteria membrane, which could be a credible barrier against antibacterial agents [47,48].” and “As we know, microorganisms have an impact on food safety during food storage. It was reported that phenolics in plants, such as grape seeds and bagasse extract, have antimicrobial properties, which might be due to degrade cell walls, disrupt protein translocation, or alter phospholipid component [49]. Based on the above results of in vitro biological activities, LSP extracts might be used as a potential natural additive to extend the shelf life of food products.” in section 3.5 to make it clearer with discussion and comparison to the literature data.
Point 6: References: check typing mistakes (uppercase letters, italics, etc.).
Response 6: Thanks for your suggestions. We have now checked and revised the typing mistakes of the references.
Round 2
Reviewer 1 Report
Comments and Suggestions for Authors
No Comments
Author Response
No response to the reviewer.
Reviewer 2 Report
Comments and Suggestions for Authors
The authors did not respond to the following queries:
Title: the title should be reformulated because the obtained extracts do not contain only phenolics and the authors cannot claim that the shown activities are related to only phenolics, thus the authors should avoid mentioning specific compounds in the title.
Introduction: In line 53, flavonoids are also antioxidant polyphenols thus they should not be separated from polyphenols; line 59, ‘’the bioactivities of phenolics’’ should be reformulated because the obtained extracts do not contain only phenolics and the authors cannot claim that the showed activities are related to only phenolics; the used plant should be explained in more detail - botanical characteristics, habitat, chemical composition, plant family, potential products on the market, etc.; the full meaning of RSM should be given; the authors should better point out the novelty of their research because it still seems that there is a lack of novelty.
Results and Discussion: the authors should explain why they used the mentioned levels of conditions (amount of solid reagent 25%, extraction time 30 min, liquid/solid ratio 20 mL/g, extraction temperature 30 °C, etc.); it is still not clear how did CCD suggest 75°C that was not used but in the case of s-to-s ratio and time, it suggested the used values; discussion is completely missed in section 2.2.1.; Table 1 - the full meaning of CCD should be given; Figure 2 - the full meaning of TPC should be given in the title although it is given in the abstract; ''TPC, especially the phenolics'' ??? but TPC is total polyphenols or phenolics; the presence of this sentence is not clear and the given explanation is not logical: ''The phenomenon of them may be attributed to the higher recovery of TPC, which was similar to the previous work.'' The presentation and discussion of the results of the antioxidant assays should be rewritten because, in this way, it is completely useless and not clear; the name of quercetin should not be in uppercase letters; it is not clear which four adding methods for LSP extract were investigated.
Material and Methods: between number and other units should be space, in line 256 and the rest of the text; the sentence in line 274 should be reformulated and the authors should find a way to not start the sentence using the number but also not use words instead of numbers; The preparation of solid reagents used in the extraction procedure should be explained in more detail; ‘’After screening through MCAE experiments’’, the preliminary screening should be explained in more detail; ‘’Based on the above result’’, which results?; the full meaning of TPC should be given in the title of Table 3 although it is given in the abstract; did the authors perform preparation of the extracts and determination of TPC in triplicate? Why does Table 3 not contain standard deviations after the values?; It is still not clear which four different methods of adding samples are used; the authors did not explain the purification and elimination of the salts from the extracts; the extracts containing these salts cannot be used in food; How can the authors say: ‘’We do not explain the purification and elimination of the salts from the extracts, because this paper does not used LSP extracts in food.’’, while the following sentence is in the manuscript: ‘’Therefore, phenolics from LSP can be considered as a potential functional food to treat diabetes by retarding the increase in blood glucose which can reduce postprandial hyperglycemia.’’ is in the manuscript, as well as numerous other sentences related to foods. Additionally, many used analyses are related to foods (such as antibacterial activity against foodborne microorganisms). Why did you make these extracts at all?; the authors should mention all positive and negative controls used in all assays for the biological activities.
Conclusion: The extraction method cannot be named green.
As far as I am concerned the manuscript still does not meet the quality criteria and impact factor of the journal Molecules.
Comments on the Quality of English LanguageEnglish language should be improved.
Author Response
Point 1: Title: the title should be reformulated because the obtained extracts do not contain only phenolics and the authors cannot claim that the shown activities are related to only phenolics, thus the authors should avoid mentioning specific compounds in the title.
Response 1: Thanks for your suggestions. We have now reformulated the title by changing “Phenolics” into “Phenolic Compounds”, and changes were made throughout the text wherever necessary. (Line 3, 14, 24, 25, 28, 31…)
Point 2: Introduction: In line 53, flavonoids are also antioxidant polyphenols thus they should not be separated from polyphenols; line 59, ‘’t’’ should be reformulated because the obtained extracts do not contain only phenolics and the authors cannot claim that the showed activities are related to only phenolics; the used plant should be explained in more detail - botanical characteristics, habitat, chemical composition, plant family, potential products on the market, etc.; the full meaning of RSM should be given; the authors should better point out the novelty of their research because it still seems that there is a lack of novelty.
Response 2: Thanks for your suggestions. We have now deleted flavonoids and changed “polyphenols [10], [11],” into “phenolic compounds [10, 11]” (Line 55) Phenolic compounds may consist of phenolics, phenolic acids, and flavonoids, so we have now changed “the bioactivities of phenolics” into “the bioactivities of phenolic compounds” (Line 60-61). We have now added the sentence “Lotus (Nelumbo nucifera Gaertn.) as a perennial aquatic plant, belonging to Nelumbonaceae family, is widely distributed and commonly cultivated in Asia and the US, which traditionally utilized for a staple food and a medicine herb in China [1].” to explain the used plant. (Line 35-37) The full meaning of RSM have been given: response surface methodology (Line 64). For the novelty of this research: Mechanochemical assisted extraction as a novel, efficient and eco-friendly technology have been widely used in preparation of bioactive components. However, the studies on the preparation of phenolic compounds are seldom rare, especially, on the preparation of phenolic compounds from lotus seedpod. This paper has used MCAE method as an innovative pre-extraction technology to extract phenolic compounds from lotus seedpod, in order to develop an efficient extraction method to recover phenolic compounds from lotus seedpod, besides, the biological activities of phenolic compounds were investigated to find an inexpensive source for natural antioxidants and the potential for functional food.
Point 3: Results and Discussion: the authors should explain why they used the mentioned levels of conditions (amount of solid reagent 25%, extraction time 30 min, liquid/solid ratio 20 mL/g, extraction temperature 30 °C, etc.); it is still not clear how did CCD suggest 75°C that was not used but in the case of s-to-s ratio and time, it suggested the used values; discussion is completely missed in section 2.2.1.; Table 1 - the full meaning of CCD should be given; Figure 2 - the full meaning of TPC should be given in the title although it is given in the abstract; ''TPC, especially the phenolics'' ??? but TPC is total polyphenols or phenolics; the presence of this sentence is not clear and the given explanation is not logical: ''The phenomenon of them may be attributed to the higher recovery of TPC, which was similar to the previous work.'' The presentation and discussion of the results of the antioxidant assays should be rewritten because, in this way, it is completely useless and not clear; the name of quercetin should not be in uppercase letters; it is not clear which four adding methods for LSP extract were investigated.
Response 3: Thanks for your suggestions. We used the mentioned levels of conditions (amount of solid reagent 25%, extraction time 30 min, liquid/solid ratio 20 mL/g, extraction temperature 30 °C, etc.) without particular reason, just because the extraction of phenolic compounds goes well under these conditions, which can be distinguished from the following optimization conditions; For the extraction temperature was set at 75°C in CCD: Actually, we did single-factor experiment of extraction temperature with the range of 30-100°C, the experiment went well when it under 90°C. But when it over 90°C, there are two phenomenons: (1) there has been consistent downward trend in TPC; (2) the extraction temperature is hard to reach 100°C, even 95°C. Therefore, taking the reality into account, we have chosen 75°C in CCD; We have now changed TPC into total phenolic content in Figure 2. (Line 142); We have now added the discussion of section 2.2.1 with three sentences: “This is because prolonged extraction time might cause the cell wall to rupture which can promote the total solubility of TPC. However, longer extraction time cannot get more TPC.”, “This might be with the increase of solution volume, mass transfer becomes more, however, continuing increase of solution volume leads to the reduce of concentration gradient [19].”, and “This phenomenon might due to the mass transfer increase under the certain threshold, once exceeded, the thermal instability of TPC occurs, leading to the degradation of TPC [20].”; We have given the full name of CCD in Table 1 as “central composite design” and the full name of TPC in Figure 2 as “total phenolic content”; We have now deleted “especially the phenolics” to make the sentence clearer; We have now rewritten the results of the antioxidant assays as “For DPPH and ABTS, the mechanism is based on hydrogen atom transfer and single electron transfer [29]. As for FRAP, the mechanism is based on single electron transfer, which was used to measure the potential of the phenolic compounds from LSP chelating Fe2+ to prevent oxidative damage [30,31]. The DPPH and ABTS assay were 279.75 18.71 µmol TE/g and 618.60 ± 2.70 µmol TE/g, respectively, which is lower than the previous work [32]. While, when its relative to FRAP, the assay was 634.14 ± 7.17 µmol TE/g, which is higher than the previous work. This phenomenon may be due to the correlation FRAP was highly correlated with phenolic acids and flavonoids [33],” (Line 197-204); We have now changed the name of quercetin in lowercase letters; For the “four adding methods for LSP extract”: I. mixed LSP extract and enzyme first and then added starch; â…¡. mixed LSP extract and starch and then added enzyme; â…¢. mixed starch and enzyme first and then added LSP extract; â…£. mixed the three together. (Line 232-234)
Point 4: Material and Methods: between number and other units should be space, in line 256 and the rest of the text; the sentence in line 274 should be reformulated and the authors should find a way to not start the sentence using the number but also not use words instead of numbers; The preparation of solid reagents used in the extraction procedure should be explained in more detail; ‘’After screening through MCAE experiments’’, the preliminary screening should be explained in more detail; ‘’Based on the above result’’, which results?; the full meaning of TPC should be given in the title of Table 3 although it is given in the abstract; did the authors perform preparation of the extracts and determination of TPC in triplicate? Why does Table 3 not contain standard deviations after the values?; It is still not clear which four different methods of adding samples are used; the authors did not explain the purification and elimination of the salts from the extracts; the extracts containing these salts cannot be used in food; How can the authors say: ‘’We do not explain the purification and elimination of the salts from the extracts, because this paper does not used LSP extracts in food.’’, while the following sentence is in the manuscript: ‘’Therefore, phenolics from LSP can be considered as a potential functional food to treat diabetes by retarding the increase in blood glucose which can reduce postprandial hyperglycemia.’’ is in the manuscript, as well as numerous other sentences related to foods. Additionally, many used analyses are related to foods (such as antibacterial activity against foodborne microorganisms). Why did you make these extracts at all?; the authors should mention all positive and negative controls used in all assays for the biological activities.
Response 4: Thanks for your suggestions. We have now added space between number and other units, and changes were made throughout the text wherever necessary; We have now reformulated “One-hundred and fifty grams stored powder” into “stored LSP powder 150 g” (Line 289) and the extraction procedure using solid reagents explained as “First, stored LSP powder 150 g was ground again by an AM400 planetary ball mill (Ants Scientific Instruments, Beijing, China); two drums and 500mL each; 10 mm stainless steel ball) at 400 r/m for 10 min; second, ground LSP power 2 g was mixed with some extent content of solid reagents (25 %, g/g) and co-grinding by the AM400 planetary ball mill with a 25mL reaction chamber and 10 mm stainless steel ball for 10 min to obtain a fine powder.” (Line 289-294); For “After screening through MCAE experiments”, we have now added “as described in section 3.2” to make it clearer (Line 309); We have now changed the sentence “Based on the above result” into “Based on the selected Li2CO3” to make it clearer (Line 310); We have now given “the full meaning of TPC” as total phenolic content in Table 3; For “did the authors perform preparation of the extracts and determination of TPC in triplicate? Why does Table 3 not contain standard deviations after the values?”, we had performed preparation of the extracts and determination of TPC in triplicate, but not for CCD in Table 3. Because the condition of the CCD experiment is based on the single factor condition that we have identified, and there is only one central condition point (50 min, 40 mL/g, and 75℃), but it will be repeated 6 times when designing the experiment, and we have done model validation in section 2.2.3. with 3 times. Thus, it had no occasion to do in triplicate for CCD; For the “four adding methods for LSP extract”: I. mixed LSP extract and enzyme first and then added starch; â…¡. mixed LSP extract and starch and then added enzyme; â…¢. mixed starch and enzyme first and then added LSP extract; â…£. mixed the three together. (Line 232-234)ï¼›For “the authors did not explain the purification and elimination of the salts from the extracts”, at first, we are so sorry to response you with “We do not explain the purification and elimination of the salts from the extracts, because this paper does not used LSP extracts in food.”, secondly, we will response it with the mechanism of MCAE, and we hope you can comprehend it clear: “The core mechanism of MCAE lies in the mechanochemical reactions during milling. Under the action of high energy mechanical force, the target components released from the broken cells react with the chemical auxiliaries, and further transferred into water-soluble or highly reactive mechanocomposites. Acid/base neutralization reaction is often used in order to improve the water-solubility of target compounds by reacting with chemical auxiliaries to generate water-soluble salts, and then dissociate target compounds by adjusting pH to achieve the effect of selective extraction and separation.” For this study, we have used glacial acetic acid to adjust the supernatant to pH 6.0 (described in Line 298-299) to recover the phenolic compounds in LSP. For “the authors should mention all positive and negative controls used in all assays for the biological activities.”, there are only negative controls used in all assays for the biological activities without positive controls. For antioxidant activities, the results were calculated based on the calibration curve of Trolox and expressed as Trolox equivalent (TE) per gram dry weight of LSP (mg TE/g DW); as for enzyme inhibitory activities, the rate of inhibition was listed in Figure 4, obviously, there are difference between the four adding methods; for antibacterial activities, the results of Figure 5 showed that when the LSP extracts diluted into 8 times there have no inhibition zone, not to mention water.
We hope that you can accept the above explanation.
